# OPENWAVES: A LARGE-SCALE ANATOMICALLY REALISTIC ULTRASOUND-CT DATASET FOR BENCHMARKING NEURAL WAVE EQUATION SOLVERS

## ABSTRACT

Accurate and efficient simulation of wave equations is crucial in computational physics, especially for wave imaging applications like ultrasound computed tomography (USCT), which reconstructs tissue properties from scattered waves. Traditional numerical solvers for wave equations are computationally intensive and often unstable, limiting their practical applications for quasi-real-time imaging. Neural operators offer an innovative approach by accelerating PDE solving using neural networks; however, their effectiveness in realistic imaging is constrained by existing datasets that oversimplify real-world complexity. In this paper, we present OpenWaves, a large-scale wave equation dataset designed to bridge the gap between theoretical equations and practical imaging applications. OpenWaves provides over 16 million frequency-domain wave simulations using real USCT configurations, featuring anatomically realistic human breast phantoms across four categories. It enables comprehensive benchmarking of popular neural operators for both forward simulation and inverse imaging tasks, allowing analysis of their performance, scalability, and generalization capabilities. By offering a realistic and extensive dataset, OpenWaves not only serves as a platform for developing innovative neural PDE solvers but also facilitates their deployment in real-world medical imaging problems.

## 1 INTRODUCTION

Imaging technology decodes wave-matter interactions and plays a critical role in scientific discoveries and biomedical diagnosis. In recent years, Ultrasound Computed Tomography (USCT) has emerged as an innovative, radiation-free method with exceptional potential for high-resolution imaging of human tissues.(Guasch et al., 2020; Li et al., 2022) As illustrated in Fig. 1(a), USCT employs a specialized transducer array—annular, cylindrical, or hemispherical—for data acquisition. Unlike conventional B-mode ultrasound, which requires manual operation and relies solely on reflected signals, USCT is fully automatic. It sequentially emits waves from each transducer and measures signals with the remaining ones, collecting both transmitted and reflected signals from tissues.(Cueto et al., 2022) This method enables USCT to reconstruct detailed 2D and 3D tissue structures similar to those produced by X-ray computed tomography (CT)(Wu et al., 2023; Zhou et al., 2023).

Wave scattering within tissues is significant in USCT because ultrasonic wavelengths are comparable to human tissue structures. To account for this, USCT employs partial differential equations (PDEs) to model wave propagation and solves a nonlinear PDE-constrained inverse problem to reconstruct tissue properties such as attenuation and sound speed.(Bernard et al., 2017; Pérez-Liva et al., 2017) This process is known as full waveform inversion (FWI). The computationally intensity and numerically instability of traditional wave equation solvers makes FWI a bottleneck for quasi-real-time USCT imaging, limiting its widespread clinical applications (Ali et al., 2024).

Neural operators have recently revolutionized PDE-based simulations and inverse problems due to their powerful approximation capabilities and fast computational speed. By leveraging neural networks to map between PDE parameter spaces and physical fields, neural operators have shown remarkable potential across diverse scientific applications, such as turbulent flow modeling, weather forecasting, and material design.(Lu et al., 2019; Li et al., 2020; 2021; Lu et al., 2021) High-quality

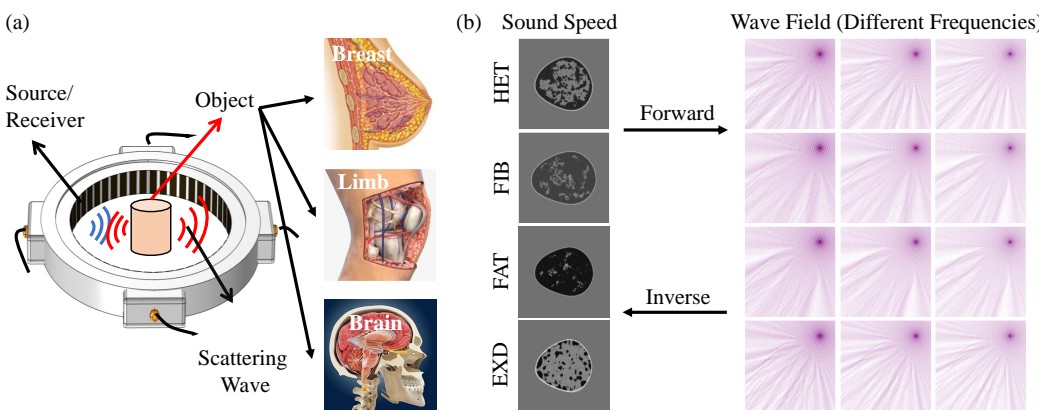

Figure 1: **Schematic diagram of a USCT system and the OpenWaves dataset.**(a) The imaging target is placed inside an annular transducer array, with each transducer emitting waves sequentially while the others act as receivers. (b) The OpenWaves dataset includes four types of anatomically realistic human breast phantoms and their corresponding wavefields at different frequencies.

PDE datasets, like PDEBench(Takamoto et al., 2022) and OpenFWI(Deng et al., 2022), have been instrumental in advancing neural operators. However, although these datasets cover various PDEs, they often simulate oversimplified scenarios—such as small regions of interest (ROIs), simple geometric boundaries, or unrealistic random porous media. These simplified settings limit the applicability to real-world problems, where complexity is much greater. To promote the practical deployment of neural operators, an application-driven, realistic, and large-scale dataset is desired.

In this paper, we introduce OpenWaves, a large-scale USCT dataset designed for benchmarking wave simulation and imaging using neural operators. OpenWaves connects theoretical wave equations with a practical medical imaging application, offering over 16 million frequency-domain wave simulations based on the Helmholtz equation (wave frequencies × source locations × scattering media → wavefields; $8 \times 256 \times 8,000 \to 16,384,000$). The dataset features anatomically realistic human breast phantoms across four categories (Fig. 1(b)), and the source locations and frequencies mimic the settings of a real annular USCT system. With its quasi-realistic setup, OpenWaves serves both as a resource for theoretical studies in deep learning and as a platform for training neural operators that can be deployed in real medical imaging systems.

We also implement multiple popular ML surrogates for both forward simulation and direct inverse imaging tasks. Both cases are evaluated using standard metrics, such as relative root mean square error (RRMSE) for forward simulation and structural similarity index measure (SSIM) and peak signal-to-noise ratio (PSNR) for imaging. Our experiments demonstrate that well-designed datasets like OpenWaves enable neural PDE solvers to enhance real-world wave simulation and imaging tasks.

In the remainder of this paper, we review related datasets and baselines (Section 2), introduce the OpenWaves dataset (Section 3), present and discuss benchmarking results (Section 4), and conclude by summarizing the dataset's contributions and limitations.

## 2 RELATED WORK

### 2.1 NEURAL OPERATORS

Neural operators are machine learning models designed to learn mappings between infinite-dimensional function spaces, enabling data-driven solutions to partial differential equations (PDEs). They are versatile tools for both forward simulations, predicting PDE solutions given parameters, and inverse problems, inferring underlying parameters from observations. For forward simulations, baseline neural operator frameworks include UNet(Ronneberger et al., 2015), which utilizes convolutional neural networks with encoder-decoder architectures; the Fourier Neural Operator (FNO)(Li et al., 2021; 2020) and its variants—UNet FNO (UFNO)(Wen et al., 2022), Born FNO (BFNO)(Zhao et al., 2023), Adaptive FNO (AFNO)(Guibas et al., 2022)—which leverage Fourier modes to capture global information efficiently; and the Multigrid Neural Operator (MgNO)(He et al., 2023), combining

multigrid methods with neural networks to handle multi-scale problems. In inverse problem, frameworks like InversionNet(Zeng et al., 2022), which uses a convolutional neural network to directly model the inversion operator, have been developed. Deep Operator Network (DeepONet)(Lu et al., 2019; 2021; Cai et al., 2021; Di Leoni et al., 2021; Lin et al., 2021) introduces a "branch and trunk" architecture to efficiently separate input functions from evaluation locations for operator learning. Fourier-DeepONet(Zhu et al., 2023) and the Neural Inverse Operator (NIO)(Molinaro et al., 2023) extend this approach by integrating DeepONet with FNO, combining local and global representations to improve accuracy and efficiency in mapping observations to PDE parameters.

## 2.2 PDE Datasets

High-quality datasets are crucial for advancing deep learning approaches to PDEs, as they provide benchmarks for training and evaluating neural operator models.(Lu et al., 2022; de Hoop et al., 2022; Benitez et al., 2023) PDEBench(Takamoto et al., 2022) is a widely used benchmark dataset that covers various forms of PDEs primarily in fluid mechanics, such as Darcy flow, advection, diffusion, and Navier-Stokes equations, but it lacks wave propagation PDEs. OpenFWI(Deng et al., 2022) specifically targets wave equations for geophysical problems, benchmarking neural networks for direct inversion from partial seismic wavefield observations. Recently, WaveBench(Liu et al., 2024) has been introduced to benchmark neural operators for forward simulations using extensive datasets of time-harmonic and time-varying wave simulations.

Despite their contributions, both OpenFWI and WaveBench assume oversimplified scattering media or sources—OpenFWI uses layered structures, and WaveBench employs Gaussian random fields and MNIST(LeCun et al., 1998) with fixed source locations—and limit simulations to small ROIs (fewer than 40 wavenumbers). These simplifications may lead to overly optimistic evaluations that fail to accurately assess neural operator performance in realistic applications, such as biomedical imaging scenarios where physical properties vary more complexly and ROIs exceed 100 wavenumbers. This underscores the need for a dataset that captures the complexities of real-world wave phenomena, motivating us to create OpenWaves, a more accurate benchmark for neural operator models in practical biomedical imaging settings. To enable consistent model comparisons, we also provide a unified PyTorch environment for benchmarking various models for forward and inverse tasks.

## 3 OpenWaves: A Realistic Application-driven Benchmark for Wave Equations

In this section, we describe the general learning problem addressed by the OpenWaves dataset, provide the detailed dataset statistics and its creation process, and discuss existing baseline models.

### 3.1 Problem Definition

The primary goal of the OpenWaves dataset is to facilitate the development of neural operators and other deep learning techniques for wave equations in real-world applications, with USCT serving as a representative example. In our dataset, we focus on steady-state (frequency-domain) wave phenomena. The propagation of ultrasonic waves is modeled by the heterogeneous Helmholtz equation, assuming negligible shear motion and nonlinear effects:

$$\left[ \nabla^2 + \left( \frac{\omega}{c(x)} \right)^2 \right] u(x) = -s(x). \tag{1}$$

Here, $\omega$ is the angular frequency of ultrasound waves, $c(x)$ is the spatial distribution of sound speed in the scattering medium, $s(x)$ is the source term, and $u(x)$ is the resulting complex acoustic field. We further assume that the variation in sound speed, $c(x)$, is confined to a pre-defined region of interest (ROI), while outside this region, the sound speed remains constant at $c_0$. This results in the following Sommerfeld radiation condition at infinity:

$$\lim_{r \to \infty} r^{\frac{n-1}{2}} \left( \frac{\partial}{\partial r} - i \frac{\omega}{c_0} \right) u(x) = 0 \tag{2}$$

Equations 1 and 2 define the relationships between $\omega$, $c(x)$, $s(x)$ and $u(x)$, which in USCT correspond to the transducer's working frequency, the properties of biological tissues, the point source located

on the annular ring, and the ultrasound wavefield, respectively. Each dataset entry consists of these four components — $\omega$, $c(x)$, $s(x)$ and $u(x)$ — allowing the dataset to support both forward wave simulation and inverse wave imaging tasks.

### 3.1.1 WAVE SIMULATION

The objective of forward wave simulation is to predict the wavefield $u(x)$ given the properties of the source $s(x)$ and the scattering medium $c(x)$. Mathematically, this task can be expressed as learning a surrogate model $\mathcal{P}: (\omega, c(x), s(x)) \rightarrow u(x;\omega)$.

Since wavefields at different frequencies exhibit distinct oscillatory behaviors, we typically train a separate deep learning model for each frequency, denoted as $\mathcal{P}_\omega : (c(x), s(x)) \rightarrow u(x)$. These models are then combined into a mixture-of-experts (MoE) framework to form the overall surrogate model, $\mathcal{P} = \{\mathcal{P}_{\omega_1}, \cdots, \mathcal{P}_{\omega_N}\}$, where $N$ represents the number of frequencies.

### 3.1.2 WAVE IMAGING

Inverse wave imaging aims to reconstruct the spatial distribution of sound speed $c(x)$ within biological tissues using the measurements from transducers. This problem is modeled as a PDE-constrained optimization:

$$\min_{c(x)} \sum_{j=1}^{N} \sum_{k=1}^{M} \left\| \boldsymbol{y}_k^j - u_k^j(\boldsymbol{x}_f) \right\|_2^2, \qquad s.t. \left[ \nabla^2 + \left( \frac{\omega_j}{c(x)} \right)^2 \right] u_k^j(x) = -s_k(x), \qquad (3)$$

where $k \in [1, M]$ indexes the transducers, $j \in [0, N]$ indexes the frequencies, $\boldsymbol{y}_k^j \in \mathbb{R}^M$ represents the measurements from all transducers when the $k$-th transducer is activated at the $j$-th frequency, and $\boldsymbol{x}_f \in \mathbb{R}^M$ denotes the transducer locations. $M$ and $N$ represent the number of transducers and frequencies, respectively. When a transducer is activated, it creates a point source $s_k(x)$. The total measurement for a given $c(x)$ forms a tensor $\mathbf{Y} \in \mathbb{C}^{M \times M \times N}$.

This inverse problem can be tackled in two ways using neural operators: 1) Gradient-based optimization: Once the forward operator $\mathcal{P}$ is learned, the image reconstruction problem becomes:

$$\min_{c(x)} \sum_{j=1}^{N} \sum_{k=1}^{M} \left\| \boldsymbol{y}_k^j - \mathcal{P}(\omega_j, c, s_k)(\boldsymbol{x}_f) \right\|_2^2 \qquad (4)$$

2) Direct inversion: Alternatively, we can approximate the inverse operator $\mathcal{P}^{-1}: (\{\omega_j\}_{j=1}^N, \{s_k\}_{k=1}^M, \mathbf{Y}) \rightarrow c(x)$ utilizing an end-to-end neural networks $\mathcal{P}_\theta^{-1}$ that directly maps the multi-frequency measurements $\mathbf{Y}$ back to $c(x)$, bypassing the need for explicit forward modeling.

## 3.2 OVERVIEW OF THE DATASET

### 3.2.1 PHYSICAL SETTINGS AND STATISTICS

OpenWaves includes 8,000 breast phantoms designed to represent the distribution of diverse human breast types in the population. As shown in Fig. 1(b), the dataset is divided into four groups, each corresponding to a specific breast density type: **heterogeneous (HET), fibroglandular (FIB), all fatty (FAT), and extremely dense (EXD)**. The wavefields are simulated using parameters from a real annular USCT system, which consists of 256 transducers arranged in a 220 mm diameter ring. The system operates at frequencies ($\omega/2\pi$) ranging from 300 kHz to 1500 kHz, corresponding to acoustic wavelengths between 1 mm and 5 mm. In our simulations, we focus on 8 frequencies between 300 kHz and 650 kHz, sampled at 50 kHz intervals, resulting in ROIs with approximately 50 to 100 wavenumbers. For each breast phantom, wavefields are simulated by activating each transducer at all frequencies, generating a total of $8,000 \times 256 \times 8 = 16,384,000$ data entries. Detailed statistics and physical settings of the dataset are summarized in Table 1.

### 3.2.2 DATA GENERATION

The dataset generation involves two key steps: 1) generating anatomically accurate breast phantoms, and 2) simulating the corresponding wavefields using real USCT system parameters.

**Phantom Generation** The breast phantoms are generated using a medical simulation tool developed by the Virtual Imaging Clinical Trial for Regulatory Evaluation (VICTRE) project at the US Food and Drug Administration (FDA).(Li et al., 2022) This tool produces 3D models of various breast anatomies, categorized into the four density types mentioned earlier. These models are sliced into 2D tissue maps, and then scaled by a random factor to simulate breasts of varying sizes. To replicate real experimental conditions, the area surrounding the breast models is filled with water.

**Wavefield Simulation** After generating the breast phantoms, we simulate the wavefields using numerical solvers based on the USCT system's source locations and frequencies. We employ the Convergent Born Series (CBS) algorithm(Osnabrugge et al., 2016), an iterative solver for simulating the Helmholtz equation. Unlike the standard Born series, CBS incorporates a preconditioner to ensure convergence, making it reliable for simulating complex media with strong scattering properties.

| Data Statistics | | | | |
|---|---|---|---|---|
| **Breast Type** | **Frequency** | **#Train/#Test** | **# Source** | **Storage** |
| Heterogeneous (HET) | 300~650 kHz | 1800/200 | 256 | 7.2TB |
| Fibroglandular (FIB) | 300~650 kHz | 2700/300 | 256 | 10.8TB |
| Fatty (FAT) | 300~650 kHz | 1800/200 | 256 | 7.2TB |
| Extremely dense (EXD) | 300~650 kHz | 900/100 | 256 | 3.6TB |
| Physical Settings | | | | |
| **Grid Spacing** | **Resolution** | **Ring Diameter** | **Source Spacing** | **Source Value** |
| 0.5 mm | $480 \times 480$ | 220 mm | $\frac{2\pi}{256}$ rad | $0.195 - 0.0275i$ |

Table 1: **Overview of OpenWaves.** Dataset composition and physical settings for data generation.

### 3.3 EXISTING BASELINES

We benchmark several existing methods for both wave simulation and wave imaging tasks on the OpenWaves dataset. All baselines are implemented in PyTorch, with detailed architectures provided in Appendix A.1. The model sizes and corresponding inference times are summarized in Table 2.

#### 3.3.1 BASELINES FOR FORWARD WAVE SIMULATION

For forward modeling, we include UNet, FNO, BFNO, AFNO, and MgNO as baseline methods:

**UNet**(Ronneberger et al., 2015) is a convolutional neural network with an encoder-decoder architecture and skip connections, effective for capturing multiscale features in images.

**Fourier Neural Operator (FNO)**(Li et al., 2021) uses Fourier transforms to parameterize integral operators, efficiently learning mappings between function spaces for solving PDEs.

**Adaptive Fourier Neural Operator (AFNO)**(Guibas et al., 2022) enhances FNO by adaptively selecting Fourier modes through an attention mechanism, improving performance on high-resolution inputs and discontinuities.

**Born Fourier Neural Operator (BFNO)**(Zhao et al., 2023) modifies FNO by incorporating the iterative Born approximation, sharing parameters across layers to better model wave scattering.

**Multigrid Neural Operator (MgNO)**(He et al., 2023) integrates multigrid techniques with neural operators for efficient and accurate modeling of multiscale phenomena.

| Forward Wave Simulation Baselines | | | Inverse Wave Imaging Baselines | | |
|---|---|---|---|---|---|
| **Model** | **# Parameters** | **Inference time [s]** | **Model** | **# Parameters** | **Inference time [s]** |
| UNet | 36.0M | 0.015 | DeepONet | 36.3M | 0.089 |
| FNO | 734M | 0.018 | InversionNet | 55.6M | 0.058 |
| AFNO | 58.6M | 0.013 | NIO | 56.3M | 0.077 |
| BFNO | 104M | 0.024 | Gradient-based | - | ~300 |
| MgNO | 26.6M | 0.015 | Optim (FNO) | | |

Table 2: **Model size and computational cost.** Comparison of the number of parameters and inference time for baseline models in both forward (Left) and inverse (Right) tasks.

### 3.3.2 BASELINES FOR INVERSE WAVE IMAGING

For inverse imaging, we benchmark DeepONet, InversionNet, and NIO for direct inversion, and also evaluate optimization-based image reconstruction using the neural operators trained for forward simulation:

**Deep Operator Network (DeepONet)**(Lu et al., 2019) employs a branch-trunk architecture to map observations to PDE parameters.

**InversionNet**(Zeng et al., 2022) proposes a CNN-based network, leveraging the exceptional capability of CNNs in handling image-related tasks.

**Neural Inverse Operator (NIO)**(Molinaro et al., 2023) combines DeepONet and FNO, with an added bagging mechanism to improve inversion accuracy and generalizability.

**Gradient-based Optimization**(Zeng et al., 2023) solves the inverse problem using conventional gradient-based methods but replaces traditional numerical wave equation solvers with the more efficient neural operators (Eq. 4).

### 3.3.3 EVALUATION METRICS

We evaluate the performance of the baseline methods independently for each task:

**Forward Modeling** Performance is measured using Relative Root Mean Square Error (RRMSE, taking mean w.r.t samples) and Maximum Error (Maximum of RRMSE w.r.t samples) across the predicted wavefields.

**Inverse Imaging** The quality of the reconstructed breast sound speed images is assessed using Structural Similarity Index Measure (SSIM) and Peak Signal-to-Noise Ratio (PSNR).

## 4 EXPERIMENTS

In this section, we present the experimental results of baseline methods on the OpenWaves dataset. Sections 4.1 and 4.2 discuss the baseline performance on forward wave simulation and inverse wave imaging tasks, respectively. In Section 4.3, we provide additional analysis on the complexities introduced by different breast types and wave frequencies in our dataset, as well as examine the scalability and generalization capabilities of the baseline models.

### 4.1 WAVE SIMULATION BENCHMARKS

We evaluated five forward simulation baselines — UNet, FNO, BFNO, AFNO, and MgNO — using a subset of OpenWaves dataset comprising wavefields at three frequencies (300, 400, and 500 kHz) from 64 uniformly sampled sources out of 256. All models were trained with relative L2 loss on four NVIDIA A800 PCIe 80 GB GPUs. Further implementation details are provided in the Appendix.

Table 3 summarizes the performance of the baselines across all breast categories and frequencies. Figure 2 presents the inference results on the test set at 300 kHz, while results for 400 and 500 kHz are provided in the Appendix A.3 (Figures 6 and 7). Quantitative analysis indicates that MgNO consistently achieved the lowest prediction errors, and all FNO variants outperformed the UNet architectures.

### 4.2 WAVE IMAGING BENCHMARKS

We compared the performance of three baselines — DeepONet, InversionNet, and NIO — and an optimization-based FWI baseline using neural operators. All methods were trained and tested using three frequencies (300, 400, and 500 kHz) data. All baselines were trained end-to-end on a single NVIDIA A800 PCIe 80 GB GPU, with measurements as input ($3 \times 256 \times 256$) and ground-truth images as output ($480 \times 480$). The optimization-based FWI performed gradient descent reconstruction, where gradients were calculated using the adjoint method (Appendix A.2) with pre-trained FNOs from Section 4.1.

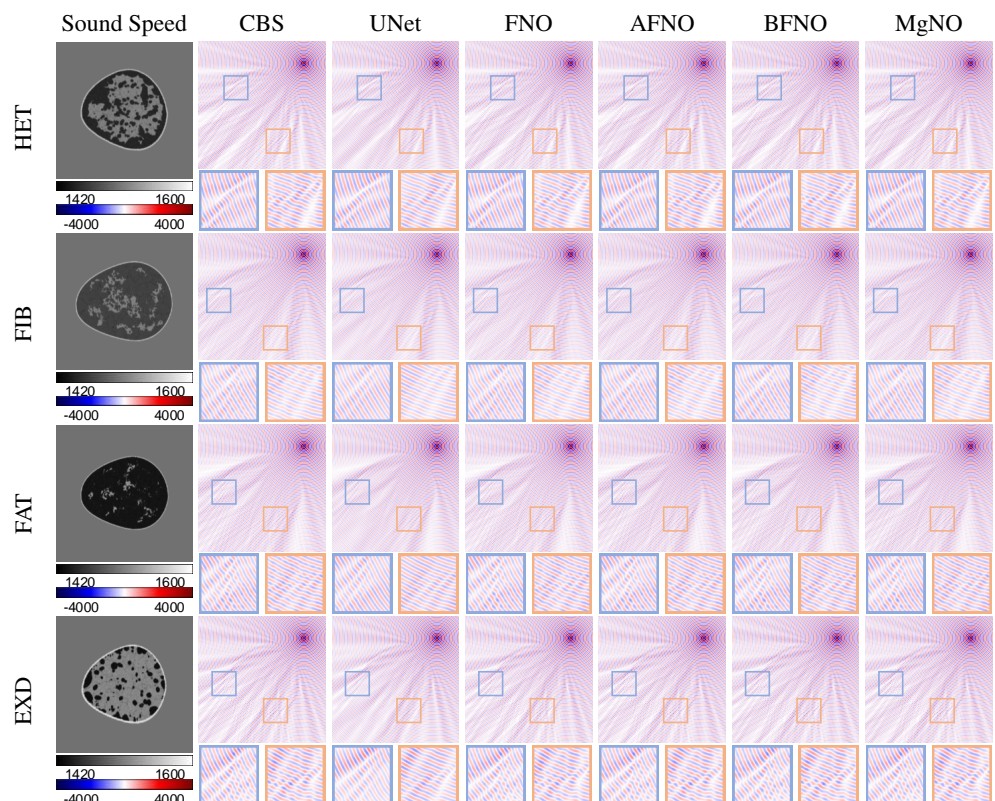

Figure 2: **Forward simulation results at 300 kHz.** Comparison of wavefield predictions for four breast types using a numerical solver (CBS) and five baseline neural operators.

| Frequency(kHz) | Metric | Models | | | | |
|---|---|---|---|---|---|---|
| | | **UNet** | **FNO** | **AFNO** | **BFNO** | **MgNO** |
| 300 | RRMSE↓ | 0.1236 | 0.0269 | 0.0165 | 0.0113 | **0.0028** |
| | Max Error↓ | 0.2551 | 0.0617 | 0.0293 | 0.0519 | **0.0092** |
| 400 | RRMSE↓ | 0.1503 | 0.0426 | 0.0242 | 0.0148 | **0.0105** |
| | Max Error↓ | 0.3017 | 0.1172 | 0.0464 | 0.0840 | **0.0244** |
| 500 | RRMSE↓ | 0.1798 | 0.0490 | 0.0276 | 0.0209 | **0.0181** |
| | Max Error↓ | 0.3571 | 0.1432 | 0.0639 | 0.0838 | **0.0410** |

Table 3: **Quantitative evaluation of forward simulation baselines.** Performance was evaluated on the test set using RRMSE and Max Error. **Bold**:Best, Underlined:Second Best

Table 4 and Figure 3 present the wave imaging performance of different methods across four breast types. Notably, NIO outperformed DeepONet on all breast categories, demonstrating the strength of the global modeling capability provided by the Fourier layer. InversionNet also achieved much better results compared to DeepONet, indicating that convolution-based networks are well-suited for complex image reconstruction tasks. It is worth mentioning that the neural operator-based optimization approach revealed significantly higher resolution than all direct inversion methods, although it incurs higher computational costs due to the iterative gradient-descent process (still much faster than traditional iterative reconstruction with numerical solvers). This suggests that the forward operators better capture the underlying wave physics, while direct inversion pipelines may overly rely on memorizing prior knowledge about the anatomy of the training breasts. In practical FWI applications, it's crucial to carefully balance reconstruction accuracy and computational efficiency.

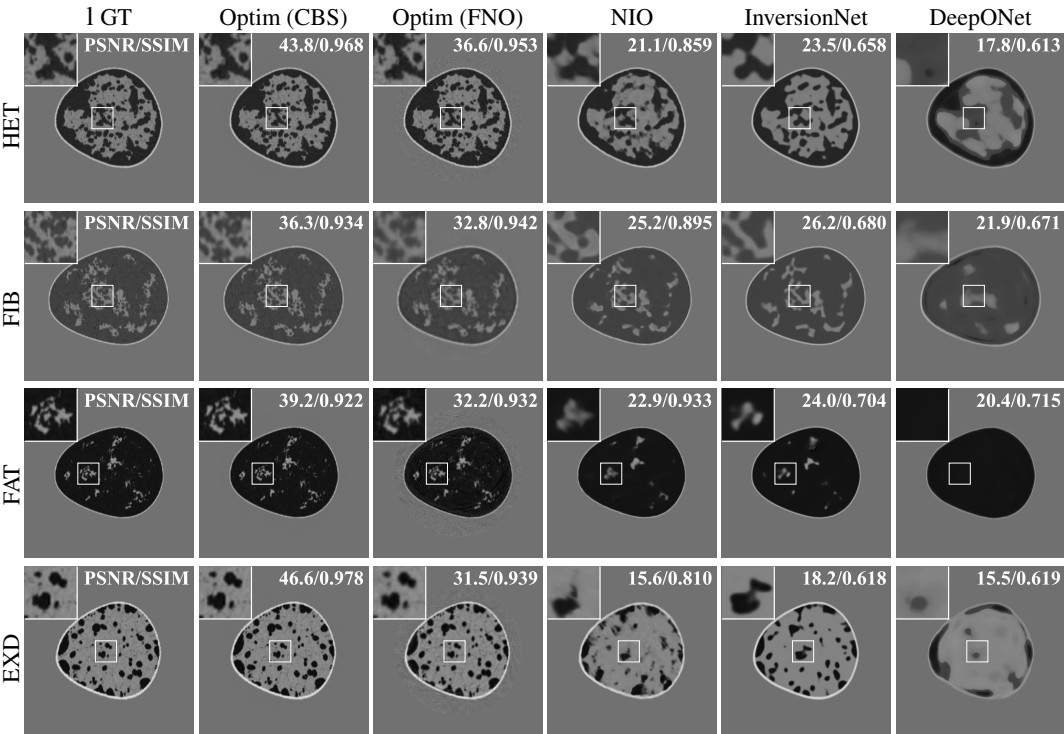

Figure 3: **Inverse imaging results.** Comparison of reconstructed breast sound speeds for four breast types using three direct inversion baselines and an optimization-based method with FNO surrogate. Results from gradient-based optimization with a numerical solver (CBS) are provided as a reference.

| Metric | Models | | | |
|---|---|---|---|---|
| | DeepONet | InversionNet | NIO | Gradient-based Optimization Method |
| PSNR↑ | 17.14 | 20.67 | 18.06 | **33.70** |
| SSIM↑ | 0.6483 | 0.6605 | 0.8680 | **0.9341** |

Table 4: **Quantitative evaluation of inverse imaging baselines.** Performance was evaluated on the test set using PSNR & SSIM. **Bold**: Best, Underlined: Second Best.

### 4.3 ADDITIONAL ANALYSIS

#### 4.3.1 DATA COMPLEXITIES

**Breast Types** Different breast categories have distinct internal structures, leading to significant variations in sound speed distribution and wave scattering effects within the tissue. As observed in Figures 1, the heterogeneous and extremely-dense breasts exhibit the most complex tissue structures and the strongest scattering because of their higher densities, while the fibroglandular and fatty breasts show the weakest scattering. This is further validated by the prediction accuracy of the learned neural operators for both forward and inverse problems as shown in Appendix A.3 (Figures 8 and 9), where heterogeneous and extremely-dense breasts reveal higher errors.

**Frequencies** All baseline neural operators experience performance degradation when learning wavefields at higher frequencies, as shown in Fig. 4 (a). This indicates that higher frequency wave equations define a more challenging operator learning task with greater complexity (Engquist & Zhao, 2018). Among all baselines, the UNet degrades the fastest, while the FNO and MgNO show less pronounced error increases. This suggests that incorporating global, local, and multiscale features is crucial for achieving high-accuracy approximations in operator learning across different frequency levels.

### 4.3.2 Scaling with Dataset Size

Figure 4 (b) examines how the performance of different forward neural operators scales with the size of the training dataset. An increased amount of training data consistently enhances wave simulation accuracy, validating the scaling law of operator learning and underscoring the necessity of creating large-scale datasets for studying neural operator frameworks. Neural operator architectures scale differently with increasing training data. Notably, MgNO and the FNO family show continued improvement as the number of training phantoms increases from 4,000 to 8,000, demonstrating better data efficiency than UNet, which shows limited improvement with additional training data.

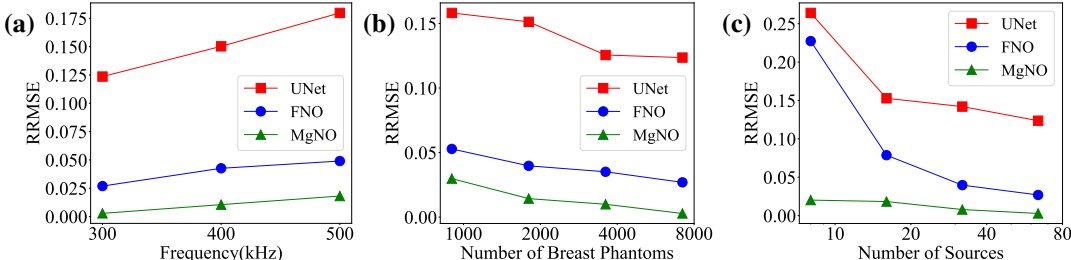

Figure 4: **Analysis of Data Complexity, Model Scalability, and Generalization.** (a) RRMSE variation of neural operators trained on data at different frequencies. (b) RRMSE variation of neural operators trained with different numbers of breast phantoms. (c) RRMSE variation of neural operators trained with different numbers of source locations.

| Metric | PSNR↑ | | | | SSIM↑ | | | |
|---|---|---|---|---|---|---|---|---|
| Train \ Test | HET | FIB | FAT | EXD | HET | FIB | FAT | EXD |
| HET | 16.07 | 12.92 | 7.50 | 10.04 | 0.8275 | 0.7311 | 0.6482 | 0.6745 |
| FIB | 12.23 | 20.13 | 9.97 | 9.57 | 0.7463 | 0.8666 | 0.8156 | 0.6390 |
| FAT | 8.16 | 9.88 | 18.35 | 6.55 | 0.7426 | 0.7739 | 0.9080 | 0.6519 |
| EXD | 12.37 | 12.92 | 8.70 | 17.58 | 0.6942 | 0.6092 | 0.6416 | **0.8402** |
| All | **19.67** | **23.71** | **21.34** | **17.89** | **0.8426** | **0.8861** | **0.9239** | 0.8339 |
| HET+FAT | 16.39 | 13.31 | 17.79 | 6.29 | 0.8320 | 0.7331 | 0.9069 | 0.6747 |

Table 5: **Quantitative evaluation of direct inversion baseline (NIO) on OOD breasts.** Each row indicates the breast type(s) used for training, and each column indicates the breast type used for testing. **Bold**: Best, Underlined: Second Best.

### 4.3.3 Generalization Capability

Previous sections demonstrated that baseline models produce strong results on in-distribution (ID) samples for both forward and inverse problems. In this section, we investigate the out-of-distribution (OOD) generalization capabilities of the representative FNO and NIO models.

**Breast Types** Figure 5 and Table 5, along with Figure 10 and Table 6 in Appendix A.3, show the performance of forward and inverse neural operators trained on selected breast types and tested across all categories. The results show that, for both forward and inverse tasks, performance on OOD samples degrades significantly compared to ID samples. However, neural operators trained on more complex breast types (e.g., heterogeneous) tend to generalize better than those trained on simpler types. Training neural operators on two significantly different breast types (e.g., heterogeneous + fatty) also enhances generalization. Additionally, the performance degradation is less pronounced in forward simulation than in inverse imaging, again suggesting that forward models better capture the underlying physics, while inverse models may tend to memorize anatomical structures.

**Source Locations** Figure 4(c) illustrates the baseline models' ability to generalize to different wave source locations. We trained the forward neural operators on datasets with varying numbers of source locations (8, 16, 32, 64) and validated them on datasets with unseen sources. As expected, prediction accuracy improves with an increasing number of training source locations. MgNO demonstrates

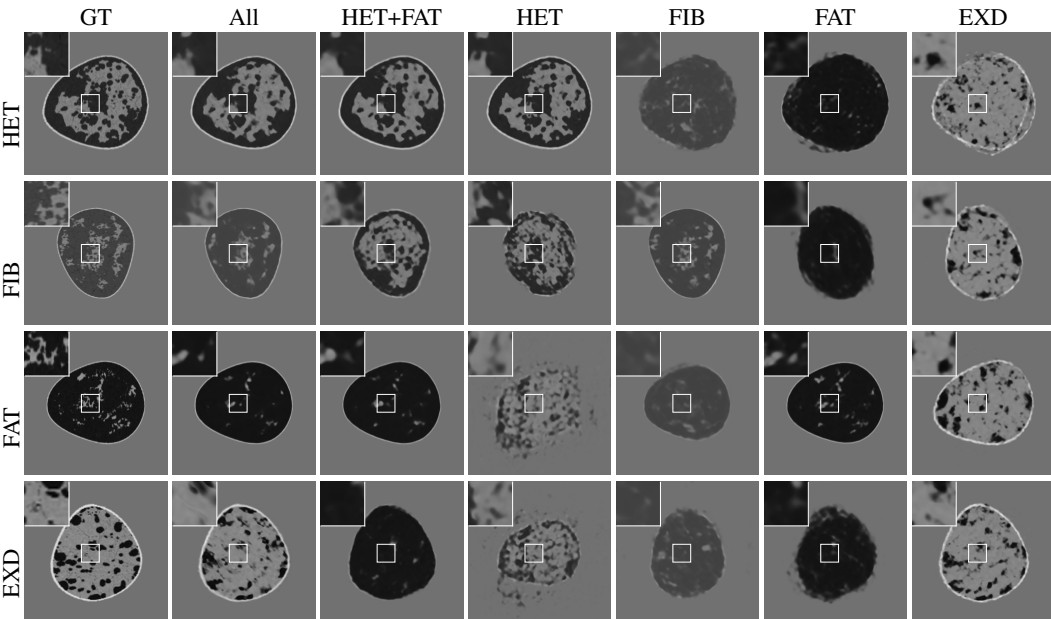

Figure 5: **Inverse imaging results of direct inversion baseline on OOD breasts.** This figure shows reconstruction samples from the NIO models. Each column represents the breast type(s) used for training, and each row represents the type used for testing. Ground-truth (GT) images are provided for reference. While direct inversion models can roughly capture the shapes of OOD samples, they tend to reproduce internal structures similar to the training data, indicating limited generalization.

strong generalization to new source locations by effectively capturing the underlying physical principles, even with limited data. As the number of sources increases, FNO's accuracy approaches that of MgNO, while UNet's performance fails to improve, indicating its difficulty in modeling wave propagation. Detailed performance for different models is provided in the Appendix A.3 (Figure 11 and Table 7).

## 5 CONCLUSION

We introduced OpenWaves, a large-scale, anatomically realistic USCT dataset designed to bridge the gap between numerical studies of wave equations and practical imaging applications. OpenWaves provides over 16 million frequency-domain wave simulations based on a real USCT system, featuring anatomically accurate human breast phantoms across four density categories. We benchmarked several baseline methods for both forward wave simulation and inverse imaging tasks, comparing their performance. Our results highlight the strengths and limitations of existing neural operator architectures, providing insights into their generalization capabilities and scalability. OpenWaves offers a valuable platform for developing and benchmarking neural wave equation solvers, enabling their application in real-world imaging tasks involving complex wave phenomena.

**Limitations** While OpenWaves represents a significant step toward realistic benchmarking of neural wave equation solvers, it has certain limitations. The dataset is currently limited to breast phantoms; including other organs like limbs or brains would enhance its applicability. Simulations are restricted to 2D due to computational constraints; incorporating 3D data would provide a more accurate representation of real-world scenarios. The dataset primarily varies sound speed as the tissue property; incorporating other properties like attenuation and anisotropy could further enhance realism. Additionally, our study focuses on neural operator architectures without extensively exploring the influence of their hyperparameters such as the number of FNO layers or other network parameters. Future work will address these limitations by expanding the dataset's scope and conducting more comprehensive analyses, aiming to provide even more valuable resources for the development of robust neural wave equation solvers.

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

## A APPENDIX

### A.1 IMPLEMENTATION DETAILS

#### A.1.1 FORWARD BASELINES

We trained all forward simulation baseline models for 30 epochs using the AdamW optimizer, with an initial learning rate of 5e-3, decayed by a StepLR scheduler (0.5 decay rate, 10-step size). We used relative L2 loss for training and RRMSE for validation. The detailed architecture of each network is provided below:

**UNet:**We implement UNet using the same structure as (Ronneberger et al., 2015) but a increased model size to other baselines for the sake of fairness. We use the UNet structure with resolution size sequence $\{[60] \times 6, [120] \times 5, [240] \times 5, [480] \times 4\}$ and 4 skip channels for Upsample block. An input block with the downsample structure using stride 1 is added to the beginning.
**FNO:**We use a vanilla FNO model with 7 FNO layers whose modes are $\{[128] \times 7\}$ and width is 40 to enlarge the representative ability.
**BFNO**: The modes and width are set to match those of FNO. Due to its parameter-sharing architecture, BFNO has a smaller parameter size compared to FNO, but its inference time is longer.
**AFNO**: The adaptive FNO uses multi-head Fourier layers that combines the attention mechanism and Fourier convolution. We set head $= 4$ and feature $= 512$ with modes list as $[40] \times 11$. The lifting operator uses Conv2d with patch size $= [4, 4]$.
**MgNO**: The model is based on the standard MgNO architecture. In this adaptation, the MGCONV modules are modified for the OpenWaves dataset by replacing the standard convolution operation with DYNAMICAL CONVOLUTION. The MgNO consists of 6 layers of MGCONV. In each MGCONV, the number of channels in each convolutional layer increases progressively as the model moves from fine to coarse levels. Specifically, the channel sizes at the five levels are $[24, 32, 64, 128, 256]$.

#### A.1.2 INVERSION BASELINES

We trained the three direct inversion baseline models for 500 epochs using the AdamW optimizer, with an initial learning rate of 1e-3 and a weight decay of 1e-6. L1 loss was used for training to preserve edges and fine details in the images, while SSIM and PSNR were used for evaluation.

**NIO:**In this paper, we modified the original setting of Convolution layer in Branch net to adapt to the resolution of this problem. For the DeepONet, a CNN with 10 Conv2d layers is applied to obtain a 512 feature coefficients and a linear layer is then applied to map it into 25 basis. The Conv2d layers we uses are listed below:

```
convblock1 = ConvBlock(1, 64, kernel_size=(1, 7), stride=(1, 2), padding
    =(0, 3))
convblock2 = ConvBlock(64, 128, kernel_size=(1, 3), stride=(1, 2),
    padding=(0, 1))
convblock3 = ConvBlock(128, 128, kernel_size=(1, 3), padding=(0, 1))
convblock4 = ConvBlock(128, 256, kernel_size=(1, 3), stride=(1, 2),
    padding=(0, 1))
convblock5 = ConvBlock(256, 256, kernel_size=(1, 3), padding=(0, 1))
convblock6= ConvBlock(256, 512, kernel_size=(1, 3), stride=(1, 2),
    padding=(0, 1))
convblock7 = ConvBlock(512, 512, kernel_size=(1, 3), padding=(0, 1))
convblock8 = ConvBlock(512, 512, kernel_size=(1, 3), stride=(1, 2),
    padding=(0, 1))
convblock9 = ConvBlock(512, 512, kernel_size=(1, 3), stride=(1, 2),
    padding=(0, 1))
convblock10 = ConvBlock(512, 512, kernel_size=(6, 4), padding = 0)
```

The trunk net uses an 8 layer MLP with 100 hidden neurons. For the FNO part, we use 4 Fourier layer with 40 modes and 32 width.
**InversionNet:**In this paper, we train the encoder and decoder of InversionNet in a supervised manner, using USCT observations from multiple sources as input and predicting 2D sound speed maps (width $\times$ height) as output. The convolution layers are adjusted to accommodate the resolution of this dataset. Additionally, in the USCT setting, we use frequency domain input structured as

Frequencies $\times$ Receiver $\times$ Source to correspond with the time domain input Source $\times$ Receiver $\times$ Time as used in seismic FWI, which improves the model's performance.

**DeepONet**:The implementation of DeepONet is the same as the DeepONet part of NIO. We further use a MLP to map the final 25 basis function to the output.

## A.2 ADJOINT METHOD IN FWI

The frequency-domain FWI can be formulated as a PDE-constrained optimization problem:

$$\min_{c(x),u_k(x)} L = \sum_{k=1}^{M} L_k = \sum_{k=1}^{M} \|y_k - u_k(x_f)\|_2^2$$

$$s.t. \left[\nabla^2 + \left(\frac{\omega}{c(x)}\right)^2\right] u_k(x) = -s_k(x). \tag{5}$$

A prevalent approach for computing the gradient, $\partial L_k/\partial c$, in FWI is the adjoint method. Using the method of Lagrange multipliers, the problem can be converted into an unconstrained form

$$\min_{c(x),u_k(x),\lambda_k(x)} \mathcal{L} = \sum_{k=1}^{M} \mathcal{L}_k = \sum_{k=1}^{M} \|y_k - u_k(x_f)\|_2^2$$

$$- \sum_{k=1}^{M} \langle \lambda_k(x), \mathcal{S}_c u_k(x) + s_k(x) \rangle \tag{6}$$

where $\mathcal{L}$ is the Lagrangian function, $\langle f, g \rangle$ denotes the real part of inner product of function $f$ and $g$ in $L^2(\mathbb{C})$, $y_k$ denotes the measurement obtained by transducer for source $s_k$, and $\mathcal{S}_c$ is the differential operator

$$\mathcal{S}_c(\cdot) = \left[\nabla^2 + \left(\frac{\omega}{c(x)}\right)^2\right](\cdot), \tag{7}$$

We then calculate the partial derivatives of $\mathcal{L}_k$ with respect to $\lambda_k, u_k, c$, respectively. Setting $\frac{\partial \mathcal{L}_k}{\partial \lambda_k}(x) = 0$ yields the Helmholtz equations itself. Similarly, enforcing $\frac{\partial \mathcal{L}_k}{\partial u_k}(x) = 0$ leads to the derivation of the adjoint equation,

$$\mathcal{S}_c \lambda_k(x) = \sum_{i=1}^{M} [u_k(x_f^{(i)}) - y_k^{(i)}] \delta(x_f^{(i)}), \tag{8}$$

where $i$ denotes the index of USCT transducers and $\delta(\cdot)$ defines a normalized point source at a specific transducer location. Substituting Eq. 7 and Eq. 8 into $\partial \mathcal{L}_k/\partial c$ results in

$$\frac{\partial \mathcal{L}_k}{\partial c}(x) = \frac{\partial \mathcal{S}_c}{\partial c}(x) \lambda_k^\star(x) u_k(x)$$

$$= -2\omega^2 \frac{\lambda_k^\star(x) u_k(x)}{c(x)^3}, \tag{9}$$

The gradient is proportional to the product of two wavefields, where $u_k(x)$ is the forward simulation result for source $s_k$ and $\lambda_k(x)$ arises from the backward simulation whose source term is defined by the discrepancies between forward predictions and measured data.

## A.3 ADDITIONAL FIGURES AND TABLES

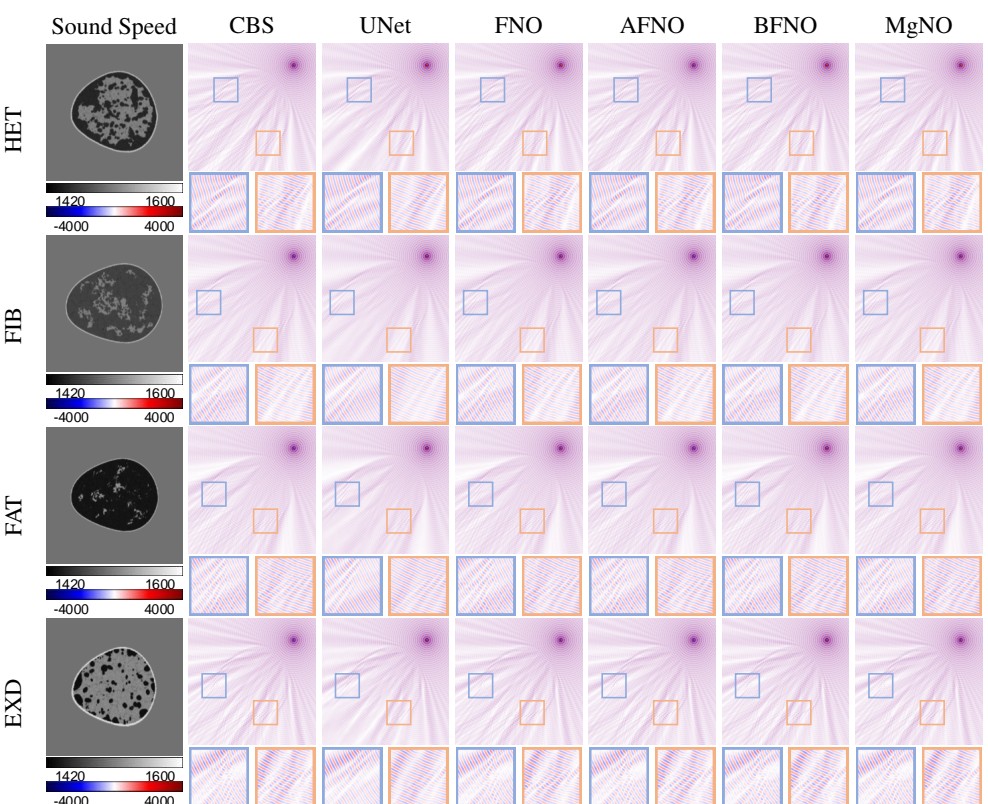

Figure 6: **Forward simulation results at 400 kHz.** Comparison of wavefield predictions for four breast types using a numerical solver (CBS) and five baseline neural operators.

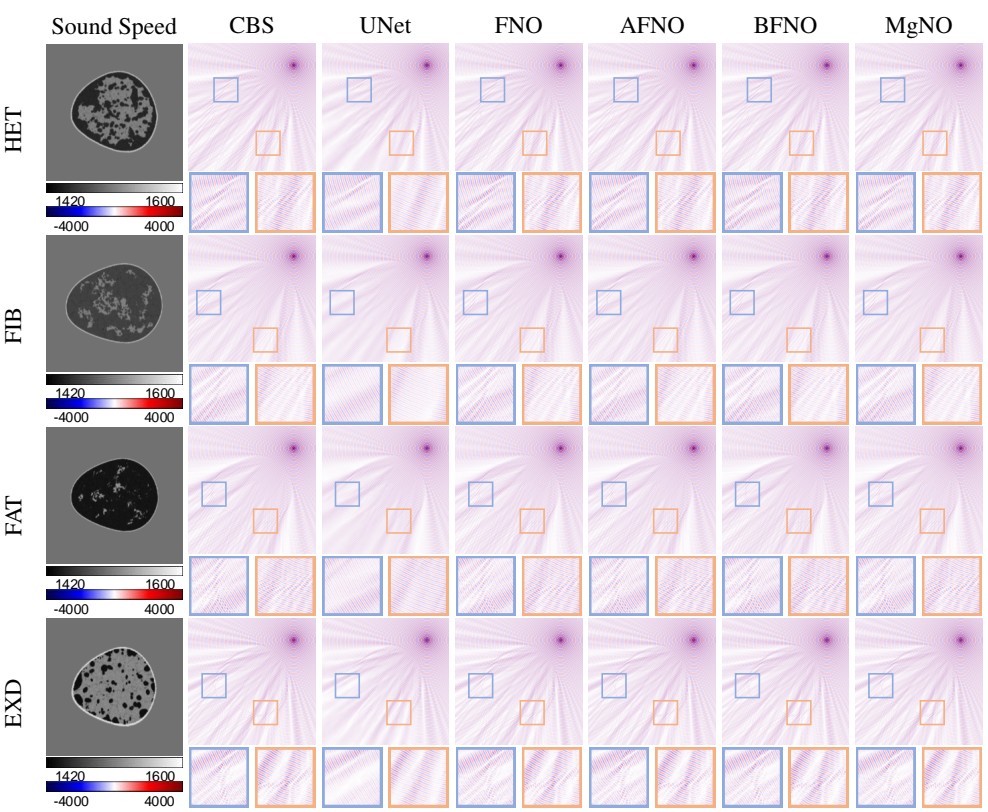

Figure 7: **Forward simulation results at 500 kHz.** Comparison of wavefield predictions for four breast types using a numerical solver (CBS) and five baseline neural operators.

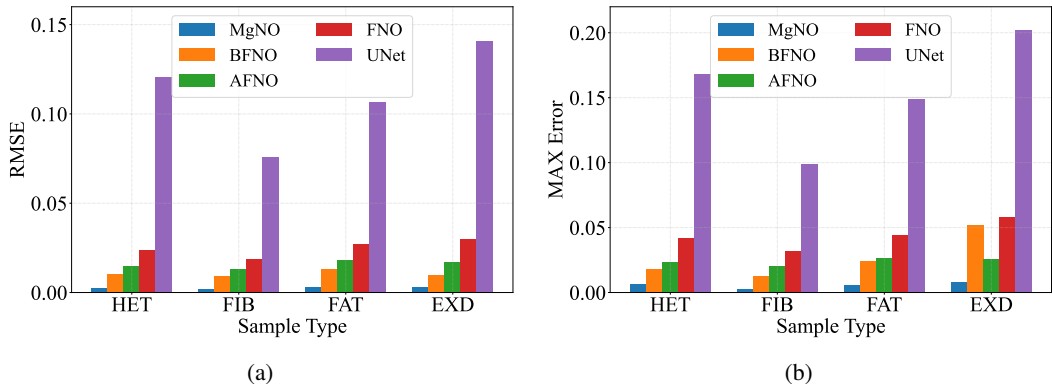

(a)

(b)

Figure 8: **Comparison of forward simulation errors across different breast categories.** RRMSE (a) and Max Errors (b) of five forward simulation baselines are reported across four breast categories. Larger errors in heterogeneous and extremely dense breasts indicate that their more complex internal tissue structures lead to stronger scattering effects and more challenging learning problems.

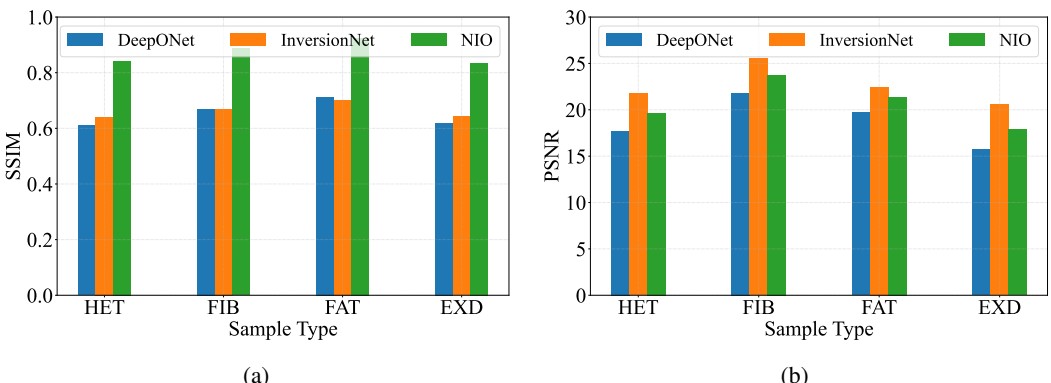

(a)

(b)

Figure 9: **Comparison of direct inversion quality across different breast categories.** SSIM (a) and PSNR (b) of three direct inversion baselines are reported for four breast categories. Lower reconstruction quality in heterogeneous and extremely dense breasts suggests that their more complex internal tissue structures lead to stronger scattering effects and more challenging learning tasks.

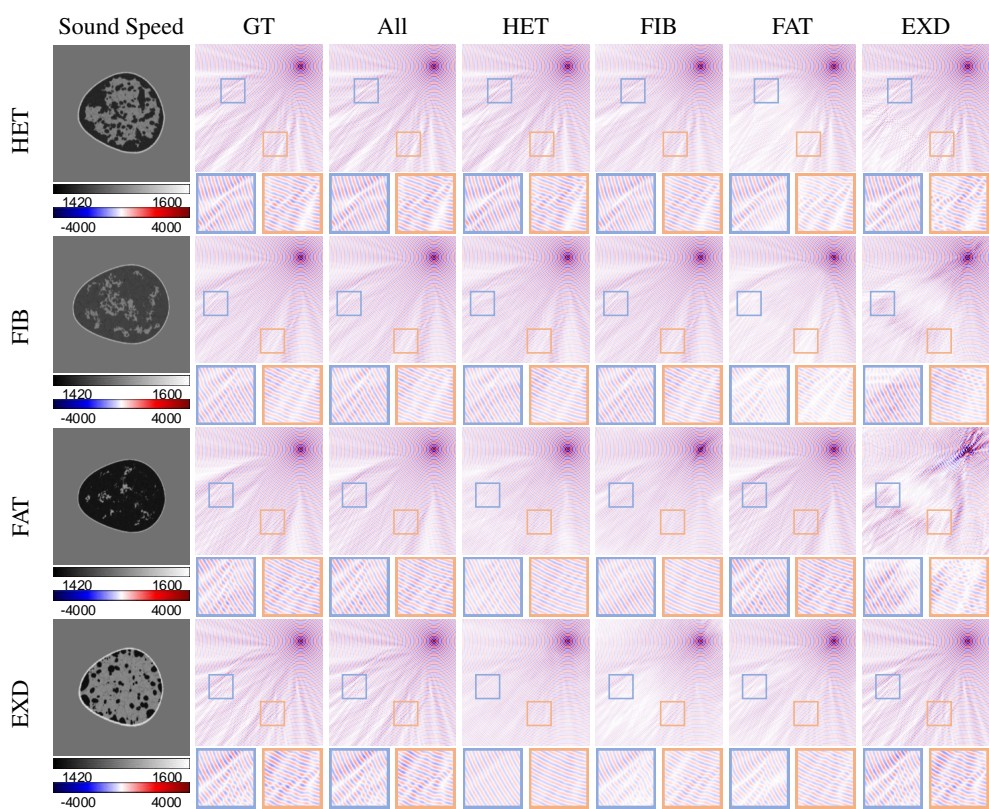

Figure 10: **Wavefield prediction results of forward simulation baseline on OOD breasts.** This figure shows wavefield prediction samples from FNO models. Each column indicates the breast type(s) used for training, and each row indicates the type used for testing. Ground-truth (GT) wavefields from the CBS solver are provided for reference. The forward simulation models demonstrate better generalization than direct inversion baselines, especially when trained on Heterogeneous (HET) and Fibroglandular (FIB) breasts. FNO trained on all four breast categories consistently achieves accurate wavefield predictions.

| Metric | RRMSE↓ | | | | Max Error↓ | | | |
|---|---|---|---|---|---|---|---|---|
| Test \ Train | **HET** | **FIB** | **FAT** | **EXD** | **HET** | **FIB** | **FAT** | **EXD** |
| HET | 0.0738 | 0.1413 | 0.8113 | 0.5210 | 0.1412 | 0.2033 | 1.0129 | 0.7653 |
| FIB | 0.2425 | 0.0208 | 0.9284 | 0.6434 | 0.4730 | 0.0523 | 1.0702 | 0.8136 |
| FAT | 0.4640 | 0.7257 | **0.0244** | 0.4966 | 0.6552 | 0.8339 | **0.0404** | 0.9889 |
| EXD | 0.2668 | 0.6802 | 1.2434 | **0.0292** | 0.5269 | 0.9783 | 2.1184 | 0.0687 |
| All | **0.0236** | **0.0187** | 0.0270 | 0.0302 | **0.0417** | **0.0318** | 0.0446 | **0.0584** |
| HET+FAT | 0.0269 | 0.5241 | 0.0287 | 0.3147 | 0.0484 | 0.7941 | 0.0545 | 0.5821 |
| FIB+EXD | 0.1918 | 0.0169 | 0.8983 | 0.0300 | 0.3753 | 0.0349 | 1.0160 | 0.0610 |

Table 6: **Quantitative evaluation of forward simulation baseline (FNO) on OOD breasts.** Each row indicates the breast type(s) used for training, and each column indicates the breast type used for testing. **Bold**: Best, Underlined: Second Best.

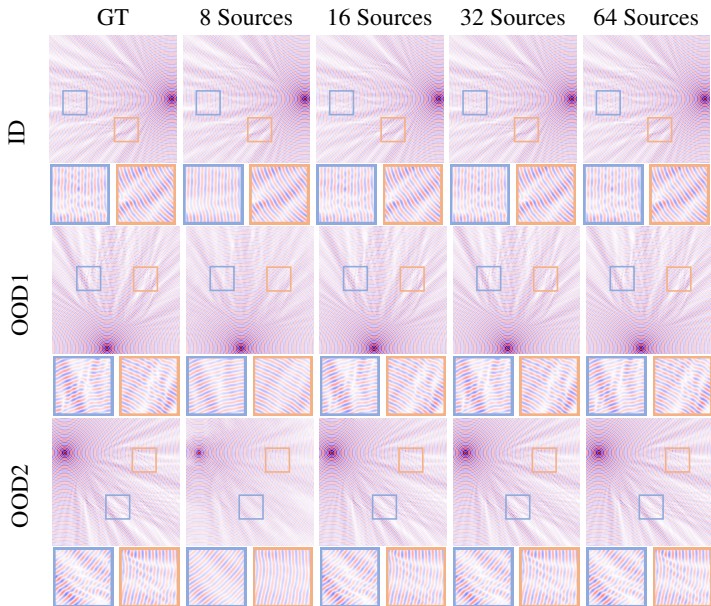

Figure 11: **Wavefield prediction results of forward simulation baseline on OOD source locations.** This figure shows wavefield predictions from FNO models trained on varying numbers of source locations. Ground-truth (GT) wavefields from the CBS solver are provided for reference. Prediction accuracy for OOD sources improves as the number of training sources increases.

| Frequency(kHz) | Metric | Models | | | | |
|---|---|---|---|---|---|---|
| | | UNet | FNO | AFNO | BFNO | MgNO |
| 300 | RRMSE↓ | 0.1237 | 0.0347 | 0.0567 | 0.0115 | **0.0041** |
| | Max Error↓ | 0.2551 | 0.0927 | 0.4447 | 0.0610 | **0.0131** |
| 400 | RRMSE↓ | 0.1532 | 0.0426 | 0.1656 | 0.0151 | **0.0108** |
| | Max Error↓ | 0.2858 | 0.1172 | 1.3172 | 0.0840 | **0.0246** |
| 500 | RRMSE↓ | 0.1877 | 0.0632 | 0.2184 | 0.0212 | **0.0183** |
| | Max Error↓ | 0.3524 | 0.1843 | 1.5160 | 0.0854 | **0.0416** |

Table 7: **Quantification of the model's generalization to OOD source locations.** Performance was evaluated by training models on the 64 source locations and testing them on the whole 256 sources (192 unseen). **Bold**:Best, Underlined:Second Best

