# OpenReview forum: "OpenWaves: A Large-Scale Anatomically Realistic Ultrasound-CT Dataset for Benchmarking Neural Wave Equation Solvers"
_ICLR.cc/2025/Conference — Submitted to ICLR 2025_

### Official Review · Reviewer_cXcq · 2024-10-31

**Soundness:** 2
**Presentation:** 2
**Contribution:** 2
**Rating:** 6
**Confidence:** 4

**Summary:**

This paper titled: OpenWaves: A Large-Scale Anatomically Realistic Ultrasound-CT Dataset for Benchmarking Neural Wave Equation Solvers presents an large-scale *simulated* dataset for USCT, including the breast image and the simulated wave imaging (from USCT). The dataset simulated 4 different breast regions, and their corresponding wave imaging. The authors thoroughly evaluated a few PDE solver for both the forward approximation and inverse problem.

**Strengths:**

1. I believe this large scale USCT dataset is valuable in terms of USCT breast related inverse problem, as it provided realistic tissue simulation.
2. I appreciate the authors for the evaluations on multiple algorithms.

**Weaknesses:**

1. From a contribution point of view, this paper is a dataset paper, however, the dataset is purely simulated and all the justifications are not convincing enough to me that this will bring important values.
a. It is perfectly fine that the paper is simulated, then the point becomes proposing a realistic USCT data generation pipeline. however, this really requires the validation from real clinical data (instead of only simulated data), and also examples with pathologies.
b. how to proof that your phantom simulation is accurate and more close to the real-breast distribution.

2. Another apparent comparison I would do is (a. training with old datasets, model 1) versus (b. trianing with new datasets, model 2), this also needs real clinical data for justification. And also ablation stuides on how the dataset makes a different, it makes very little sense to train on this distribution and evaluate on the same distribution.

**Questions:**

1. Based on the creation of this dataset, do you have any insights of how to design a better PDE solver specifically for this USCT application.

---

> ### Author Response · Authors · 2024-11-20
>
> \
> We sincerely thank the reviewer for their constructive suggestions and are pleased that they found our dataset “valuable in terms of USCT breast-related inverse problems” and appreciated our “evaluations on multiple algorithms.” Below, we address the concerns raised:
> \
> \
> **1. Validation on real clinical data**
>
> &ensp;&ensp;To validate the dataset under real-world conditions, we applied a surrogate model trained on OpenWaves to reconstruct a public USCT dataset of real breasts with tumors [1]. The imaging results, provided in the new supplementary PDF, demonstrate that models trained on simulated data can produce high-quality reconstructions of real-world data. This serves as empirical evidence that the generated phantoms closely resemble the anatomy of real organs. To further support the research community, we will include these public experimental samples in the open-source dataset, encouraging researchers to test and evaluate their own models.
> \
> \
> **2. Training and validation on different datasets**
>
> &ensp;&ensp;Unfortunately, there are no other publicly available USCT datasets for training or testing. OpenWaves is the first dataset that incorporates realistic anatomy and USCT instrument settings. Prior datasets, such as OpenFWI[2] and WaveBench[3], assume unrealistic scenarios with overly simple scattering media (e.g. layered structures, Gaussian random fields) and confined computational domains (<40 wavenumbers). These limitations make them unsuitable for addressing realistic USCT imaging problems.
> However, we think we partially address this concern through experiments in Section 4.3.3, where neural operators trained on specific breast types or source locations are tested on unseen types and locations, demonstrating the model’s generalization to out-of-distribution (OOD) data. Additionally, the successful application of OpenWaves-trained models to real clinical data, as mentioned before, highlights the strong alignment between the simulated and real data distributions.
> \
> \
> **3. Insights on neural PDE designs**
>
> &ensp;&ensp;Our benchmarking experiments provided multiple valuable insights into designing neural PDE solvers. For instance, multi-grid architectures, which effectively handle high-frequency oscillations, and BFNO, which incorporates multi-scattering processes in its design, outperform standard Fourier layers used in FNO for wave simulations and imaging. These findings suggest that integrating physical priors into neural PDE architectures is key to improving performance in USCT imaging.
> For inverse problems, we found that forward operators generalize better than direct inverse mappings trained on the same dataset. This suggests that forward simulation is inherently easier to learn, making the combination of forward neural operators with adjoint optimization a more robust and efficient solution for wave imaging. We will explicitly summarize and highlight these insights in the updated version of the paper.
> \
> \
> We sincerely thank the reviewer again for their valuable insights and welcome further questions or discussions. Thank you!
> \
> \
> **Reference:**
>
> [1] Rehman Ali et al. 2-D Slicewise Waveform Inversion of Sound Speed and Acoustic Attenuation for Ring Array Ultrasound Tomography Based on a Block LU Solver. IEEE Transactions on Medical Imaging. 2024
>
> [2] Chengyuan Deng et al. OpenFWI: Large-scale multi-structural benchmark datasets for full waveform inversion. NeurIPS 2022.
>
> [3] Tianlin Liu et al. WaveBench: Benchmarking Data-driven Solvers for Linear Wave Propagation PDEs. TMLR 2024.

---

> > ### Comment · Reviewer_cXcq · 2024-11-25
> > **regarding real clinical results**
> >
> > I appreciate the authors for their feedback.
> > THe results on real clinical dataset is very interesting, i'm not an expert in this field, do you mind adding some ablation studies on these real-clinical datasets? just qualitatively is fine.

---

> > > ### Author Response · Authors · 2024-11-26
> > >
> > > Thank you for your suggestions and interest! We’ve added more ablation studies in the updated supplementary material (.zip). It includes reconstructions of clinical breast data using different neural networks (two forward neural operators and one direct inversion network) and evaluations under various settings, such as different numbers of receivers and frequencies.

---

> > > > ### Comment · Reviewer_cXcq · 2024-11-27
> > > >
> > > > Great, i appreciate the new results, i truly think the quality of the paper has been improved a lot.
> > > > I would increase my score, but i still think the realy clincal study should be a big part of the main paper,
> > > > thats said, a good simulation dataset is valuable "only if" it shows promising results on real clinical dataset and achieve SOTA result compared to other methods, in terms of both experiments and ablation studies.

---

> > > > > ### Author Response · Authors · 2024-11-28
> > > > >
> > > > > Thank you! We really appreciate your valuable feedback that has greatly improved the paper. We will include a thorough real clinical study in our camera-ready version.

---

### Official Review · Reviewer_6sNf · 2024-11-03

**Soundness:** 2
**Presentation:** 3
**Contribution:** 1
**Rating:** 3
**Confidence:** 4

**Summary:**

The work introduced the dataset OpenWaves, which provides 16M frequency-domain wave simulations on breast phantom. The dataset can help to evaluate the forward simulation and tomography inversion in ultrasound tomography.

**Strengths:**

1. The authors have conducted 16 million simulations across a substantial number of breast phantoms, requiring significant computational and storage resources. If proven to be high-fidelity, this dataset could save future researchers considerable time and effort in data preparation.

2. The authors provide a thorough comparison of different models in both forward simulation and backward reconstruction, offering a relatively comprehensive evaluation.

**Weaknesses:**

1. Lack of original contribution: This work does not appear to introduce any novel methods in dataset construction for ultrasound tomography. Creating such a dataset fundamentally depends on two aspects: the simulation algorithm and the generation or collection of breast phantoms. However, in this study, both are adopted from existing resources—the VICTRE project for phantom generation and the Convergent Born Series (CBS) algorithm for wave field simulation. Consequently, this dataset could theoretically be reproduced by others with sufficient computational resources, given VICTRE and CBS, without any unique contribution from the authors.

2. Insufficient discussion on parameter selection and real-world distribution: VICTRE requires a range of carefully chosen parameters to generate realistic breast phantoms, such as surface shape, internal compartmentalization, etc. However, this work does not discuss how parameters in OpenWave were selected or how their distributions might mirror real-world variability. Understanding these choices is essential for evaluating the fidelity of the benchmark and whether it can accurately assess the performance of different frameworks.

3. Lack of quality control measures: Although VICTRE enables simulation of breast phantoms, these simulations inherently lack the complexities of real-world samples. The authors do not address how they ensured the quality or realism of the generated phantoms. For instance, was there any subsampling or external validation from clinical experts to assess the fidelity of these simulations? Such quality assessment is crucial for establishing the dataset as a reliable benchmark.

4. Lack of support of breast type: What is the rationale behind the distribution of four breast types?1000:3000:2000:2000. Can authors provide the literature that supports the figure?

**Questions:**

1. The current citation format is structured as though it serves as the subject of a sentence. To improve clarity and consistency, the authors should either use \citep or add parentheses to the current citations.

2. The authors frequently reference the 16M frequency-domain wave simulations to emphasize the dataset's large scale. However, this may be misleading. If my understanding is correct, each breast phantom has 256 sources, and during reconstruction, data from all sources would typically be required. I suggest that the authors represent the dataset size as either 8000 or 8000x8, which would provide a more accurate and meaningful metric for researchers.

---

> ### Author Response · Authors · 2024-11-20
> **Response to Reviewer 6sNf (Part1)**
>
> \
> We sincerely thank the reviewer for the thoughtful feedback and for recognizing that our dataset “could save future researchers considerable time and effort in data preparation” and that our evaluation is “relatively comprehensive.” Below, we address each of the reviewer’s concerns in detail.
> \
> \
> **1. Original contribution**
>
> &ensp;&ensp;OpenWaves is the first wave PDE dataset to simulate realistic scenarios with complex scattering media and large wavenumbers, bridging the gap between existing datasets and real-world problems. It addresses a critical issue in the ML-PDE community: models over-optimized for toy settings often fail to generalize to practical challenges.
> \
> \
> &ensp;&ensp;OpenWaves also goes beyond merely using existing simulation tools: we designed a pipeline that automatically assigns accurate sound speeds to tissues, fine-tuned the hyperparameters of numerical solvers, and designed realistic USCT boundary conditions for robust USCT simulation. Additionally, the paper provides a thorough evaluation of forward and inverse neural PDE baselines, offering insights for designing effective solvers. As with prior benchmark works [1, 2], this processes of data generation, standardization, validation, and benchmarking should be recognized as significant contributions to the field.
> \
> \
> **2. Parameter selection for realistic breast generation**
>
> &ensp;&ensp;We indeed carefully chose the VICTRE parameters and are happy to discuss them below.
> \
> \
> &ensp;&ensp;To ensure shape diversity and anatomical realism, we adjusted five key parameters—a1b, a1t, a2l, a2r, and a3—that control the breast’s bottom, top, left, right, and outward scales, respectively (units in cm). These were sampled from truncated Gaussian distributions:
> a1b, a1t, a2l, a2r$\sim\mathcal{TN}(5.0, 2.0, 3.5, 7.5)$, and a3/a1b$\sim\mathcal{TN}(1.4, 0.1, 1.0, 1.5)$, with $\sim\mathcal{TN}(\mu,\sigma,𝑎,𝑏)$ representing a truncated Gaussian distribution in the interval (a, b).
> \
> \
> &ensp;&ensp;To model the internal structure, we first adjusted the targetFatFrac parameter to control fat distribution, as it mainly determines the division of four breast types. Typically, the targetFatFrac parameter of Extremely Dense, Heterogeneous, Fibroglandular, and Fatty types is respectively in range of (0, 0.25), (0.25, 0.5), (0.5, 0.75), and (0.75, 1.0). We also fine-tuned the backFatBufferFrac parameter in range of (0, 0.01) to force a little fraction of phantom in nipple direction to be fat. To define skin properties, we mainly adjusted the following parameters: SkinScale in range of (200,400), SkinScaleNippleDir in range of (5,20), and skinStrength in range of (0.5,2.0).
> \
> \
> &ensp;&ensp;We will include these additional details about the data generation process in the  Appendix in camera-ready version.
> \
> \
> **3. Quality assessment of the generated phantoms**
>
> &ensp;&ensp;We agree that assessing the quality of the generated phantoms is essential for establishing the dataset as a reliable benchmark. In the newly uploaded supplementary PDF, we report a new experiment where a neural surrogate model trained on OpenWaves effectively reconstructs a public clinical dataset of human breasts with tumors[3]. These results empirically validate that OpenWaves captures key complexities of real-world samples. With medical doctors on our team, we plan to include additional assessments, such as external validation from clinical experts, in the revised manuscript.
> \
> \
> **4. Distribution of breast types**
>
> &ensp;&ensp;We apologize for not clearly explaining the rationale for the breast type distribution. According to [4], the ratio of Heterogeneous, Fibroglandular, Fatty, and Extremely Dense types is [28%, 40%, 23%, 9%]. We adjusted this slightly to account for the higher breast density in Asian populations. Additionally, we found a labeling error where the counts for Heterogeneous and Extremely Dense breasts were swapped. This has been fixed in the revised version.
> \
> \
> **Reference:**
>
> [1] Chengyuan Deng et al. OpenFWI: Large-scale multi-structural benchmark datasets for full waveform inversion. NeurIPS 2022.
>
> [2] Tianlin Liu et al. WaveBench: Benchmarking Data-driven Solvers for Linear Wave Propagation PDEs. TMLR 2024.
>
> [3] Rehman Ali et al. 2-D Slicewise Waveform Inversion of Sound Speed and Acoustic Attenuation for Ring Array Ultrasound Tomography Based on a Block LU Solver. IEEE Transactions on Medical Imaging. 2024
>
> [4] Li F, Villa U, Park S, et al. 3-D stochastic numerical breast phantoms for enabling virtual imaging trials of ultrasound computed tomography[J]. IEEE transactions on ultrasonics, ferroelectrics, and frequency control, 2021, 69(1): 135-146.

---

> ### Author Response · Authors · 2024-11-20
> **Response to Reviewer 6sNf (Part2)**
>
> \
> \
> **5. Citation format**
>
> &ensp;&ensp;We apologize for the oversight in citation format. This issue has been corrected in the revised version.
> \
> \
> **6. Interpretation of the dataset’s size**
>
> &ensp;&ensp;The interpretation of the dataset’s size depends on its intended use. For forward simulations, each (source, phantom → wavefield) combination constitutes a data pair, resulting in 16M items. For inverse problems, as the reviewer noted, the dataset includes measurements corresponding to 8000 breast phantoms. However, as discussed in Section 4.2, training a neural surrogate for forward simulations is a more effective approach, as it captures accurate physics and supports more precise gradient-based image reconstruction. Directly learning inverse mappings often overfits to common structures in the phantoms rather than capturing the underlying scattering physics, leading to biased reconstructions. This is why we emphasize the 16M data pairs in this context, but we are happy to rephrase this for clarity.
> \
> \
> We sincerely thank the reviewer again for the valuable suggestions and are happy to address any additional questions. Thank you!

---

> ### Author Response · Authors · 2024-11-28
>
> Dear reviewer, we hope you’ve had a chance to review our rebuttal and the additional experiments (Supplementary Materials, zip file) we provided in response to your comments. Given only five days left in the rebuttal period, if there are any remaining concerns or questions, we’d be happy to clarify or provide further results. Please let us know if our response has resolved your concerns.

---

> ### Author Response · Authors · 2024-12-01
>
> Dear Reviewer,
> \
> \
> With only one day left in the rebuttal period, we kindly and urgently request your review of our rebuttal and the additional experiments provided in the Supplementary Materials (zip file) in response to your comments. If you have any remaining concerns or questions, we would be more than happy to clarify them or provide further results as needed. However, a lack of response at this stage would significantly impact the fairness of the evaluation process for our work.
> \
> \
> Thank you!

---

### Official Review · Reviewer_eLgk · 2024-11-04

**Soundness:** 3
**Presentation:** 4
**Contribution:** 3
**Rating:** 6
**Confidence:** 3

**Summary:**

This paper presents a new dataset for ultrasound computed tomography (USCT) that: 1) establishes a connection between a theoretical wave equation and a practical medical imaging application; 2) simulates data based on the Helmholtz equation; and 3) conducts large-scale benchmarking of neural operator and end-to-end baselines.

**Strengths:**

The paper provides a comprehensive evaluation of the simulated data, including experiments on various organs of interest, different imaging models, and various baselines. In addition, the presentation of the paper is clear and easy to follow.

**Weaknesses:**

1) My primary concern pertains to the dataset generation. Despite a thorough review of the introduction and background, I find it unclear why generating this new dataset is particularly challenging. From my understanding, this dataset is generated purely through simulation. As long as one implements the wave equation according to the theoretical framework and generates the digital phantom via simulation tool, reproducing this dataset seems straightforward. From this perspective, the main contribution of this dataset appears to be its potential to save researchers time in regenerating these terabyte-scale data. Could the authors clarify this point?

2) Furthermore, the differences between this dataset and existing datasets are not sufficiently discussed. While the manuscript notes that other datasets employ different imaging models, it would strengthen the work if the authors also discussed additional differences, such as scale, imaging applications, or computational complexity. Comments on the applicability of these datasets in comparison to the proposed dataset would also be beneficial.

3) Since this paper focuses on medical applications of USCT, a purely simulated dataset (both the ground truth and measurements) may be seen as less important, particularly given that medical professionals often prioritize real data. I could not locate the evaluation criteria specifically for the dataset and benchmark track for ICLR. I am concerned whether such a purely simulated dataset meets ICLR's requirements.

**Questions:**

1) Could the authors provide insights into the time complexity involved in generating the entire proposed dataset?

---

> ### Author Response · Authors · 2024-11-20
>
> \
> We sincerely thank the reviewer for their valuable feedback. We are pleased that the reviewer found our paper to “provide a comprehensive evaluation of data” with “clear presentation.” Below, we address each of the reviewer’s concerns in detail.
> \
> \
> **1. Main contribution**
>
>  &ensp;&ensp;OpenWaves is the first wave PDE dataset designed to simulate realistic scenarios with complex scattering media and large wavenumbers, addressing a critical gap in the ML-PDE community. Existing datasets often oversimplify problems by assuming small wavenumbers and basic scattering structures (e.g., layered geometries, Gaussian random fields), resulting in models that overfit toy settings and fail to generalize to real-world challenges. OpenWaves addresses this gap by introducing real-world-like complexity and conducting extensive evaluations of forward and inverse neural PDE baselines, providing valuable insights into solver design, as detailed in our summarized rebuttal to all reviewers.
> \
> \
>  &ensp;&ensp;Regarding data generation, simulated datasets are standard in the neural PDE community and align with the practices of ICLR and other prestigious AI conferences. As with prior works [1, 2, 3], the processes of data generation, standardization, validation, and benchmarking are recognized as significant contributions to the field. Furthermore, OpenWaves also goes beyond merely using existing simulation tools: we designed a pipeline that automatically assigns accurate sound speeds to tissues, selected appropriate numerical solvers, fine-tuned their hyperparameters, and designed realistic USCT boundary conditions for robust USCT simulation.
> \
> \
> **2. Differences from existing datasets**
>
>  &ensp;&ensp;As discussed in Sec. 2.2 of the original paper, OpenWaves differs significantly from prior datasets like OpenFWI and WaveBench. OpenWaves provides anatomically realistic breast phantoms, real-world USCT imaging parameters, complex scattering media, and high wavenumbers. Additionally, it is more than an order of magnitude larger than these datasets, enabling studies of highly oscillatory PDEs, data efficiency, and scaling properties of neural architectures. Key differences are summarized in the following table:
>
> |  | &ensp;&ensp;&ensp; &ensp;&ensp;&ensp;Application |&ensp;&ensp;&ensp; &ensp;&ensp;&ensp;&ensp;&ensp;Scattering Media Properties | Data Pairs | Wavenumbers |  &ensp;&ensp; &ensp;&ensp;&ensp;&ensp;&ensp;&ensp;&ensp;&ensp;Numerical Solver | File Size |
> |:---:|:---:|:---:|:---:|:---:|:---:|:---:|
> | OpenWaves | Ultrasound CT | anatomically-realistic breasts  | 16,384k  | [48, 104] | Convergent Born Series  | 28.8TB  |
> | OpenFWI[1] | Geophysical Imaging |  Layered structures or style-transferred natural images  | 1,396k  | [2, 10] | Finite difference method  | 2.1TB  |
> | WaveBench[2] | Abstract PDE Inverse Problem |  Gaussian random fields, MNIST  | 2,463k  | [3, 34] |  hybridizable discontinuous Galerkin (HDG) method | 88GB  |
>
> \
> **3. Validation on real clinical data**
>
>  &ensp;&ensp;We performed a new experiment in the new supplementary PDF to validate the dataset. A surrogate model trained on OpenWaves was applied to reconstruct a public USCT dataset of real breasts with tumors [4]. The imaging results show high-quality reconstructions, demonstrating that OpenWaves-generated phantoms closely resemble the anatomy of real organs. These results will be added to the revised manuscript. We will also highlight this public dataset in our open-source release to enable researchers to test and evaluate their own models.
> \
> \
> **4. Time complexity for dataset generation**
>
>  &ensp;&ensp;The dataset generation involves two main stages:
>
> 1) Breast Phantom Generation: High-resolution breast phantoms using VICTRE took approximately 40 hours on an Intel i7-13700KF CPU, excluding time spent tuning hyperparameters.
>
> 2) Wave Simulation: Wavefield simulations for the entire dataset required approximately 60 hours of computation, parallelized across 8 NVIDIA A800 PCIe 80 GB GPUs and 20 CPUs. This estimation also neglects the additional time spent fine-tuning solver hyperparameters.
>
>  &ensp;&ensp;We will include this information in our revised manuscript.
> \
> \
> We sincerely thank the reviewer again for the constructive suggestions and are happy to address any further questions or discussions. Thank you!
> \
> \
> **Reference:**
>
> [1] Chengyuan Deng et al. OpenFWI: Large-scale multi-structural benchmark datasets for full waveform inversion. NeurIPS 2022.
>
> [2] Tianlin Liu et al. WaveBench: Benchmarking Data-driven Solvers for Linear Wave Propagation PDEs. TMLR 2024.
>
> [3] Makoto Takamoto et al. PDEBench: An extensive benchmark for scientific machine learning. NeurIPS 2022.
>
> [4] Rehman Ali et al. 2-D Slicewise Waveform Inversion of Sound Speed and Acoustic Attenuation for Ring Array Ultrasound
>
> Tomography Based on a Block LU Solver. IEEE Transactions on Medical Imaging. 2024

---

> > ### Comment · Reviewer_eLgk · 2024-11-25
> >
> > Thank you to the authors for their response. However, I remain skeptical about treating data simulation as a main contribution in a machine learning conference. Shouldn't establishing a robust and clean pipeline for data generation or simulation be considered a fundamental responsibility of a researcher?
> >
> > Additionally, while the authors mentioned that experiments on real data were conducted, I could not find the corresponding results in their response.
> >
> > Therefore, I will maintain my score.

---

> > > ### Comment · Reviewer_cXcq · 2024-11-25
> > > **regarding the real data results**
> > >
> > > I checked the supplementary zip file, and the correspoinding results are there :)

---

> > > > ### Comment · Reviewer_eLgk · 2024-11-26
> > > >
> > > > Thanks! I didn't check the supplementary zip file as the main pdf fils has the appendix section already.
> > > >
> > > > I agree with Reviewer cXcq. The experiments on the real clinical data is really interesting, and more qualitative results could be helpful.

---

> > > > > ### Author Response · Authors · 2024-11-26
> > > > >
> > > > > Thank you for your suggestions and interest! We’ve added more ablation studies in the updated supplementary material (.zip). It includes reconstructions of clinical breast data using different neural networks (two forward neural operators and one direct inversion network) and evaluations under various settings, such as different numbers of receivers and frequencies.

---

> > > > > ### Author Response · Authors · 2024-11-28
> > > > >
> > > > > Thank you once again for taking the time to review our paper. With only five days remaining in the rebuttal period, we hope you’ve had a chance to review our response and the additional experiments provided in the Supplementary Materials (zip file). If you have any remaining concerns, questions, or need further clarification or additional results, we’d be more than happy to assist.

---

> > ### Author Response · Authors · 2024-12-01
> >
> > Dear Reviewer,
> > \
> > \
> > With only one day left in the rebuttal period, we kindly and urgently request your review of our rebuttal and the additional experiments provided in the Supplementary Materials (zip file) in response to your comments. If you have any remaining concerns or questions, we would be more than happy to clarify them or provide further results as needed. However, a lack of response at this stage would significantly impact the fairness of the evaluation process for our work.
> > \
> > \
> > Thank you!

---

### Official Review · Reviewer_kTeM · 2024-11-04

**Soundness:** 3
**Presentation:** 2
**Contribution:** 3
**Rating:** 5
**Confidence:** 3

**Summary:**

This paper introduces a large-scale dataset named OpenWaves, which provides over 16 million frequency-domain wave simulations based on real ultrasound-CT configurations. The dataset is specifically designed to include anatomically realistic human breast phantoms.

**Strengths:**

1. Benchmarking wave simulation and imaging for neural operators represents a significant contribution to the field.

2. The dataset is extensive, with over 16 million simulations, covering a variety of anatomically realistic human breast phantoms.

**Weaknesses:**

1. The mathematical formulation in Section 3.1 requires improvement. Beyond simply explaining variable meanings, the authors should define variables in greater detail, including their range and space. For instance, the symbol $\nabla$ in Eq. 1 is not clearly defined, and the partial derivatives in Eq. 2 are also ambiguous.

2. Although Section 4.3.2 discusses the scaling property, the dataset's efficiency remains somewhat unclear. For example, as noted by the authors, the performance of different neural operators varies with dataset size, yet the underlying rationale and its implications for designing the proposed dataset require clarification.

**Questions:**

1. Do the authors plan to open-source the curated dataset?

2. Have the authors considered comparing the simulated dataset to a smaller real-world dataset to assess the alignment of data distributions with real-world conditions?

3. Could the authors improve the spacing of references?

---

> ### Author Response · Authors · 2024-11-20
>
> \
> We thank the reviewer for the thoughtful feedback and for recognizing our dataset as “extensive,” “anatomically realistic,” and “a significant contribution to the field.” Below we provide further explanations and additional results to address each concern raised by the reviewer.
> \
> \
> **1. Mathematical formulation**
>
> &ensp;&ensp;In Section 3.1, we only define the general wave equation without specifying variable ranges. The variables’ quantitative values are explained in Section. 3.2.1 of the original paper: the region of interest (ROI) of $u(x)$  is 220mm by 220mm,  $\frac{\omega}{2\pi}$ spans 300 kHz to 650 kHz, and the sources $s(x)$ are positioned on a annular ring, etc.
> Equation 1 presents the standard form of the Helmholtz equation, where
> $\nabla^2$ represents the Laplace operator$\nabla^2{u}= \frac{\partial^2{u}}{\partial{x^2}}+\frac{\partial^2{u}}{\partial{y^2}}$.
> Equation 2 specifies the Sommerfeld radiation condition as the standard boundary condition, ensuring the solution’s uniqueness. Here, $r = \Vert{\mathbf{x}}\Vert$ is the radial distance from the source and $n=2$ represents the spatial dimension.
> The partial derivative in Cartesian coordinates is defined as:
> $\frac{\partial{u(\mathbf{x})}}{\partial{r}}=\frac{\mathbf{x}_1}{r}\frac{\partial{u(\mathbf{x})}}{\partial{x_1}}+\frac{\mathbf{x}_2}{r}\frac{\partial{u(\mathbf{x})}}{\partial{x_2}}$.
> We will include additional explanations about these variables and operators in the final version to reduce ambiguity.
> \
> \
> **2. Scaling property and data efficiency**
>
> &ensp;&ensp;The scaling behavior of a model depends on its expressive power. The UNet’s high loss levels indicate its inability to capture high-frequency features, reflecting a limitation in the model itself, not the dataset’s efficiency. By contrast, more expressive models like MgNO and FNO achieve lower loss values as the dataset size increases, demonstrating that OpenWaves effectively evaluates diverse baselines’ performance and scaling properties.
> \
> \
> **3. Open-sourcing the dataset**
>
> &ensp;&ensp;As a dataset and benchmark paper, we will make both the dataset and baseline code publicly available upon acceptance.
> \
> \
> **4. Alignment of data distributions**
>
> &ensp;&ensp;To validate the alignment between OpenWaves dataset and real-world conditions, we used a surrogate model trained on OpenWaves to reconstruct real breast USCT data[1]. The imaging results, provided in the new supplementary PDF, show high-quality breast reconstructions, demonstrating a strong consistency between the simulated and real-world data distributions.
> \
> \
> **5. Spacing of references**
>
> &ensp;&ensp;We apologize for the formatting oversight in the citation spacing. This has been corrected in the revised version.
> \
> \
> We sincerely thank the reviewer again for the valuable feedback and are happy to address any additional questions or concerns. Thank you!
> \
> \
> **Reference:**
>
> [1] Rehman Ali et al. 2-D Slicewise Waveform Inversion of Sound Speed and Acoustic Attenuation for Ring Array Ultrasound Tomography Based on a Block LU Solver. IEEE Transactions on Medical Imaging. 2024

---

> ### Author Response · Authors · 2024-11-28
>
> Thank you again for your efforts in reviewing our paper. Given only five days remaining in the rebuttal period, we kindly remind you to review our response and additional experiments (Supplementary Materials, zip file), and let us know if we have satisfactorily addressed your concerns. If you have any remaining questions or require further clarification or additional results, we’d be happy to provide them.

---

> ### Author Response · Authors · 2024-12-01
>
> Dear Reviewer,
> \
> \
> With only one day left in the rebuttal period, we kindly and urgently request your review of our rebuttal and the additional experiments provided in the Supplementary Materials (zip file) in response to your comments. If you have any remaining concerns or questions, we would be more than happy to clarify them or provide further results as needed. However, a lack of response at this stage would significantly impact the fairness of the evaluation process for our work.
> \
> \
> Thank you!

---

> > ### Comment · Reviewer_kTeM · 2024-12-01
> >
> > Thank you to the authors for providing a detailed rebuttal. After carefully considering the additional clarifications in the updated supplementary materials and reviewing the comments from other reviewers, I have decided to maintain my original score.

---

> > > ### Author Response · Authors · 2024-12-02
> > >
> > > Thank you for your response. We believe we have addressed all your concerns and would greatly appreciate it if you could let us know which specific points remain unresolved. We are happy to provide further clarifications or improvements if needed. A template response like this is not helpful.

---

### Author Response · Authors · 2024-11-20

\
We thank the reviewers for their constructive feedback and for recognizing the significance of our work, including the importance of the dataset to the field (kTeM, cXcq), its extensive scale (kTeM, 6sNf), comprehensive evaluation (eLgk, 6sNf, cXcq), and clear presentation (eLgk). Below, we address main concerns raised in the reviews. Point-to-point responses are included as a reply to each reviewer, and we also provide additional experimental results in the newly submitted supplementary PDF.
\
\
**1. The key contributions of our work**

&ensp;&ensp;OpenWaves is the first large-scale wave PDE dataset to simulate realistic scenarios with complex scattering media and large wavenumbers, addressing a critical gap in existing datasets. Current datasets oversimplify the problem by assuming small wavenumbers and basic scattering media structures (e.g., layered geometries or Gaussian random fields), resulting in a long-standing yet overlooked issue in the machine learning for PDE community: over-optimistic evaluations and biased models that perform well in toy settings but fail to generalize to real-world challenges.[1] OpenWaves focuses on realistic medical imaging applications, particularly ultrasound computed tomography (USCT) for breast disease diagnosis, offering a benchmark that promotes the development of neural PDE solvers tailored to practical challenges. This dataset is valuable for both the machine learning and medical imaging communities.

&ensp;&ensp;Our work also provides a thorough evaluation of forward and inverse neural PDE baselines, offering insights for designing effective solvers. For instance, multi-grid frameworks are particularly effective at learning high-frequency oscillations, and combining forward neural operators with adjoint optimization demonstrates superior generalization to unseen data compared to direct inverse mapping. These findings provide actionable guidance for future research.
\
\
**2. Data generation using existing tools**

&ensp;&ensp;Utilizing existing simulation tools for data generation is a common practice when building datasets. However, as a dataset & benchmark paper, the processes of generation, standardization, validation, and benchmarking should be recognized as significant contributions to the field, as in prior works [2, 3, 4]. Moreover, OpenWaves goes beyond merely using existing simulation tools: Since VICTRE was originally designed for X-ray mammography, we developed pipelines to assign anatomically accurate sound speed to tissues, modeling realistic acoustic properties of scattering media. Additionally, we incorporated USCT-specific experimental setups into numerical solvers, enhancing the dataset's realism and relevance.
\
\
**3. Generalization to real-world data evaluation**

&ensp;&ensp;Neural PDE solvers trained on OpenWaves generalize well to real-world data. Preliminary results on a public clinical dataset of human breast[5] is included in the supplementary material, confirming our dataset's anatomical fidelity. More detailed quantitative evaluations will be provided in the final version.

&ensp;&ensp;We hope these clarifications and additional results address your concerns. We are happy to discuss any remaining points during the discussion phase. Thank you!
\
\
**Reference:**

[1] Nick McGreivy, Ammar Hakim. Weak baselines and reporting biases lead to overoptimism in machine learning for fluid-related partial differential equations. Nature Machine Intelligence 2024.

[2] Makoto Takamoto et al. PDEBench: An extensive benchmark for scientific machine learning. NeurIPS 2022.

[3] Chengyuan Deng et al. OpenFWI: Large-scale multi-structural benchmark datasets for full waveform inversion. NeurIPS 2022.

[4] Tianlin Liu et al. WaveBench: Benchmarking Data-driven Solvers for Linear Wave Propagation PDEs. TMLR 2024.

[5] Rehman Ali et al. 2-D Slicewise Waveform Inversion of Sound Speed and Acoustic Attenuation for Ring Array Ultrasound Tomography Based on a Block LU Solver. IEEE Transactions on Medical Imaging. 2024

---

### Author Response · Authors · 2024-12-03
**Rebuttal Summary**

### **Dear Reviewers, ACs, and PCs,**

We sincerely thank all the reviewers for their valuable and constructive feedback, as well as for dedicating their time to reviewing our paper. Based on the insightful suggestions provided during the rebuttal phase, we have conducted a thorough revision addressing the reviewers' key concerns. Below, we summarize these concerns and detail the revisions and updates included in the final submission. We hope this provides the reviewers, ACs, and PCs with a clearer understanding of the progress and outcomes of the rebuttal discussion.

---

### **Concerns**
- **[kTeM]** Mathematical formulation of Ultrasound CT imaging problem.
  - **[Authors]** We add additional clarifications in the response. *[Response to kTeM]*
- **[kTeM]** Scaling property and data efficiency.
  - **[Authors]** We explain in the response. *[Response to kTeM]*
- **[kTeM]** Open Sourcing the dataset.
  - **[Authors]** We explain in the response. *[Response to kTeM]*
- **[eLgk, 6sNf, cXcq]** Main Contribution of Dataset and Benchmark.
  - **[Authors]** We explain in the response. *[Response to eLgk, 6sNf, cXcq] [General Response: The key contributions of our work] [General Response: Data generation using existing tools]*
  - **[Authors]** We conduct additional experiments to demonstrate the usefulness of the proposed dataset in real clinical dataset imaging. *[Supplementary Materials]*
- **[eLgk, cXcq]** Differences from existing datasets.
  - **[Authors]** We explain in the response. *[Response to eLgk, cXcq] [General Response: The key contributions of our work]*
- **[kTeM, eLgk, 6sNf, cXcq]** Alignment of real-world data.
  - **[Authors]** We explain in the response. *[Response to kTeM, eLgk, 6sNf, cXcq] [General Response: Generalization to real-world data evaluation]*
  - **[Authors]** We conducted additional experiments to evaluate the generalization capabilities of different models on clinical datasets. *[Supplementary Materials]*
- **[eLgk]** Time complexity of dataset generation.
  - **[Authors]** We explain in the response. *[Response to eLgk]*
- **[6sNf]** Parameter selection for dataset generation.
  - **[Authors]** We explain in the response. *[Response to 6sNf]*
- **[6sNf]** Quality assessment of generated phantoms.
  - **[Authors]** We explain in the response. *[Response to 6sNf]*
- **[6sNf]** Distribution of breast types.
  - **[Authors]** We provide reference support in the response. *[Response to 6sNf]*
- **[cXcq]** Insights on neural PDE designs.
  - We explain in the response. *[Response to cXcq] [General Response: The key contributions of our work]*

---

### **Revisions**
- **[Overview of the dataset]** We fix a labeling error of the breast type according to concerns raised by Reviewer 6sNf. *[L 227-236, Section3.2.1]*
- **[Supplementary Materials]** We add validation experiments on using different models for real clinical breast datasets.
  - **Reconstruction results:** We successfully reconstruct a clear malignant tumor together with other internal tissue structures, using BFNO with 8 frequencies and 256 transmitters.
  - **Ablation study from different models:** We test another two models, (a) FNO with 8 frequencies and 256 transmitters, and (b) NIO inversion.
  - **Ablation study from different experimental settings:** We employ three observational data settings about frequency and source number, including (a) 8 frequencies and 128 transmitters, (b) 8 frequencies and 64 transmitters, and (c) 3 frequencies and 128 transmitters.

---

### **Other Points**
- **[kTeM]** Spacing of references
  - **[Author]** Citations misused. *[Corrected]*

---

### **Summary**
We sincerely thank the reviewers for their valuable suggestions, which have helped us strengthen our revised submission. We believe the revised submission could be much more robust and contribute to broadening the field of neural PDE and medical imaging, offering valuable insights to the community. We deeply appreciate the efforts of the reviewers, ACs, and PCs during the rebuttal period.

*Best regards,*

*Authors*

---

### Author Response · Authors · 2024-12-04

Dear ACs,

Thank you for your efforts in overseeing the review process for our paper. We deeply appreciate the valuable feedback and constructive criticism provided by the reviewers, which have significantly contributed to improving our work.

However, during the rebuttal phase, we observed that both Reviewer kTeM and Reviewer 6sNf did not actively engage in meaningful discussions or address the clarifications and responses we provided. This lack of engagement may limit the extent to which their reviews accurately reflect the quality of our submission.

In light of this, we sincerely request that you take our responses into account and reconsider the value and weight of these two reviewers' comments in the decision-making process.

Sincerely,

Authors

---

### Meta-Review · Area_Chair_TCKd · 2024-12-22

**Metareview:**

This paper describes a large dataset of realistic breast phantoms with varying material properties and corresponding high-frequency ultrasound CT measurements. The dataset is designed to develop and benchmark neural operators for solving the associated forward and inverse problems. Most reviewers agreed that this new curated dataset could help push neural operators for wave inversion into clinically relevant territory (where they are currently not). I agree with that. Two reviewers were more critical, and unfortunately several unresolved concerns remain which I believe are important. This is why I am recommending a (weak) rejection (but I would not mind if the paper gets accepted). I think that the addressing the following points would greatly help in the resubmission:

- A clear up-front argument for the importance and uniqueness of OpenWaves and its potential to unlock neural operators for breast cancer screening. There is now a comparison in 2.2 but I recommend moving this very early and maybe evening having a paragraph titles "Why is this dataset needed to make progress on ...". It would be even better if the authors could demo some of the overoptimistic evaluations resulting from earlier datasets; this may not be too hard
- A better delineation in 4.3.2 about what is meant by scaling. Scaling laws for models and data are currently a white-hot research area thanks to LLMs, foundation models, and all that. The related questions revolve around the optimal co-design of datasets and models, and has been the intention of kTeM. Perhaps the authors' intention was to simply show that more data = better performance, scaled by model size, but this is actually an interesting question. At any rate, a delineation of what is meant would be strategically helpful.
- More details about quality control in the main text. In particular, does the breast phantom capture sufficient variation to ensure neural operators which don't remove or hallucinate features? It is great that the authors ran a test on public clinical data, but that should be mentioned prominently in the main text, and there should be a corresponding quantitative / qualitative evaluation at least for some samples (also a comparison to a strong principled iterative inversion). While some neural operator designs may generalize well even if trained of "poor" data (of low variation), some that would perform strongly may need high quality data. A discussion would strengthen the paper.

I hope that the authors can effectively. use these suggestions and the discussions to revise and resubmit their paper.

**Additional Comments On Reviewer Discussion:**

eLgk was wondering about suitability of dataset papers for ICLR (they are welcome!) and qualitative evaluation.  kTeM asked about math and scaling. 6sNf was concerned about existing datasets and novelty, parameter selection, quality of the generated phantoms (similarity to actual breast). cXcq wondered about suitability of simulated dataset papers (they are welcome!), realism of phantoms, clinical justification. The authors addressed some of the concerns during rebuttal but overall no strong support emerged for the paper in its current form and the critical reviews remained critical.

It is important to highlight that the response to rebuttal by kTeM had no useful content or pointers to unresolved concerns which would help guide the authors. Even more unfortunate is that despite reminders 6sNF did not respond at all. While both of them did contribute to the post-rebuttal discussion, I wish they had acted more conscientiously during the rebuttal phase.

---

### Decision · Program_Chairs · 2025-01-22

Reject